# Pairing statistics and melting of random DNA oligomers: Finding your partner in superdiverse environments

**Simone Di Leo**[1☉], **Stefano Marni**[1☉], **Carlos A. Plata**[2☉], **Tommaso P. Fraccia**[3], **Gregory P. Smith**[4], **Amos Maritan**[5], **Samir Suweis**[5], **Tommaso Bellini**[1]*

**1** Dipartimento di Biotecnologie Mediche e Medicina Traslazionale, Università degli Studi di Milano, Milano, Italy, **2** Física Teórica, Universidad de Sevilla, Sevilla, Spain, **3** Institut Pierre-Gilles de Gennes, CBI UMR 8231, ESPCI Paris, Université PSL, CNRS, Paris, France, **4** Department of Physics and Soft Materials Research Center, University of Colorado, Boulder, Colorado, United States, **5** Dipartimento di Fisica 'G. Galilei', INFN, Università di Padova, Padova, Italy

☉ These authors contributed equally to this work.
* tommaso.bellini@unimi.it

**Data Availability Statement:** All relevant data are within the manuscript and its Supporting information files.

## Abstract

Understanding of the pairing statistics in solutions populated by a large number of distinct solute species with mutual interactions is a challenging topic, relevant in modeling the complexity of real biological systems. Here we describe, both experimentally and theoretically, the formation of duplexes in a solution of random-sequence DNA (rsDNA) oligomers of length $L$ = 8, 12, 20 nucleotides. rsDNA solutions are formed by $4^L$ distinct molecular species, leading to a variety of pairing motifs that depend on sequence complementarity and range from strongly bound, fully paired defectless helices to weakly interacting mismatched duplexes. Experiments and theory coherently combine revealing a hybridization statistics characterized by a prevalence of partially defected duplexes, with a distribution of type and number of pairing errors that depends on temperature. We find that despite the enormous multitude of inter-strand interactions, defectless duplexes are formed, involving a fraction up to 15% of the rsDNA chains at the lowest temperatures. Experiments and theory are limited here to equilibrium conditions.

## Author summary

Several biological processes require that specific partner molecules succeed in binding after negotiating their way through a huge number of interactions with other molecules. How such molecular recognition emerges among millions distinct molecular species is an open problem. We have studied, both experimentally and theoretically, such process of "molecular recognition" in pools of highly diverse random DNA oligomers, which binds preferentially, but not exclusively, to its perfect complementary sequence. We find a complex behavior, in which some perfect pairing takes place with a non-trivial temperature dependence that we understand thorough statistical mechanics modelling. The pairing pattern of short random DNA is relevant in the context of the origin of life since the so-

**Funding:** S.D.L., S.M. and T.B. acknowledge support from MIUR-PRIN (Grant No. 2017Z55KCW). T.P.F. acknowledges IPGG Laboratoire d'Excellence, "Investissement d'avenir" program ANR-10-IDEX-0001-02 PSL, ANR-10-LABX-31 and ANR-10-EQPX-34. S.S. acknowledges UNIPD grant BIRD209912. C.A.P. acknowledges the support from PGC2018-093998-B-I00 funded by FEDER/Ministerio de Ciencia e Innovaciòn-Agencia Estatal de Investigaciòn (Spain) and from the program PAIDI-DOCTOR by Junta de Andalucìa and European Social Fund. The funders had no role in study design, data collection and analysis, decision to publish, or preparation of the manuscript.

**Competing interests:** The authors have declared that no competing interests exist.

called "RNA World" was most probably based on the mutual recognition of random chains.

## Introduction

One of the defining features of biomolecules is the specificity and selectivity of their mutual interactions. Selectivity is at the heart of virtually all biological processes, including cell signalling, immune response, genetic transmission and regulation of gene expression. These processes are based on the presence of partner molecules that succeed in finding and docking to each other after negotiating their way through a huge number of collisions and interactions with other molecules, some of which exert attraction [1]. Indeed, in all actual cases, specific biomolecular interactions take place in "superdiverse" environments, i.e. in contexts of enormous variety of molecular species. Succeeding in pairing to the target is thus depending not only on the binding strength between partner molecules, but on the whole network of pair interactions between concurring molecular species, possible cooperativity, concentration and degeneracy. While the complexity of interactions in the contexts of biomolecular crowding is universally acknowledged [2, 3], the thermodynamics and statistical physics of large pools of distinct interacting molecules have been discussed only in the frame of phase transitions [4, 5], and molecular models of such systems have not been presented yet.

DNA oligonucleotides mixtures are one of the pre-eminent systems to experimentally recreate the conditions described above in a controlled way, given the natural selectivity of base pairing [6] and the possibility to artificially synthesize large ensembles of distinct DNA sequences with controlled distributions [7]. Herein, we consider systems formed by aqueous solutions of DNA oligonucleotides of length $L$ in which the four bases (Cytosine, C; Guanine, G; Adenine, A; Thymine, T) are present with equal probability in each position in the sequence as sketched in Fig 1a. Thus, in each of these random-sequence DNA oligomers (rsDNA) solutions, $4^L$ distinct molecules can be found with approximately the same probability. We consider solutions of rsDNA oligomers with $L$ = 8, 12 and 20 (8$N$, 12$N$ and 20$N$), corresponding to mixtures of $\approx 7 \cdot 10^4$, $2 \cdot 10^7$ and $10^{12}$ distinct sequences, respectively; in these systems we study interaction selectivity, defect distribution and equilibration properties as a function of the temperature.

Because of its relevance, the hybridization of nucleic acids has been a topic of continuous investigation since their identification, which led to well-established tools and models to calculate free energy, melting temperature and secondary structure directly from the DNA (or RNA) sequences involved [8–11]. In particular, the thermodynamics of duplex formation is commonly evaluated in the frame of the so-called "Nearest Neighbor" (NN) model, in which the hybridization free energy is obtained by the summation of elemental contributions. These are extracted from database of melting temperatures of DNA oligomers and effectively take into account both WC and non-canonical pairings [8]. However, available models are accurate only in the case of pools of low numbers of distinct sequences [12], and do not have the capability of predicting the behavior of complex systems such as the rsDNA solutions considered here.

In a rsDNA solution, upon colliding, pairs of rsDNA molecules bind to each other with a strength and resultant stability that is mainly determined by the level of complementarity of the base sequences according to the Watson-Crick (WC) pairing rules. This results in an interaction matrix, sketched in Fig 1b, where each position represents the most stable duplex conformation for a given pair. The most probable outcome of a random collision between two

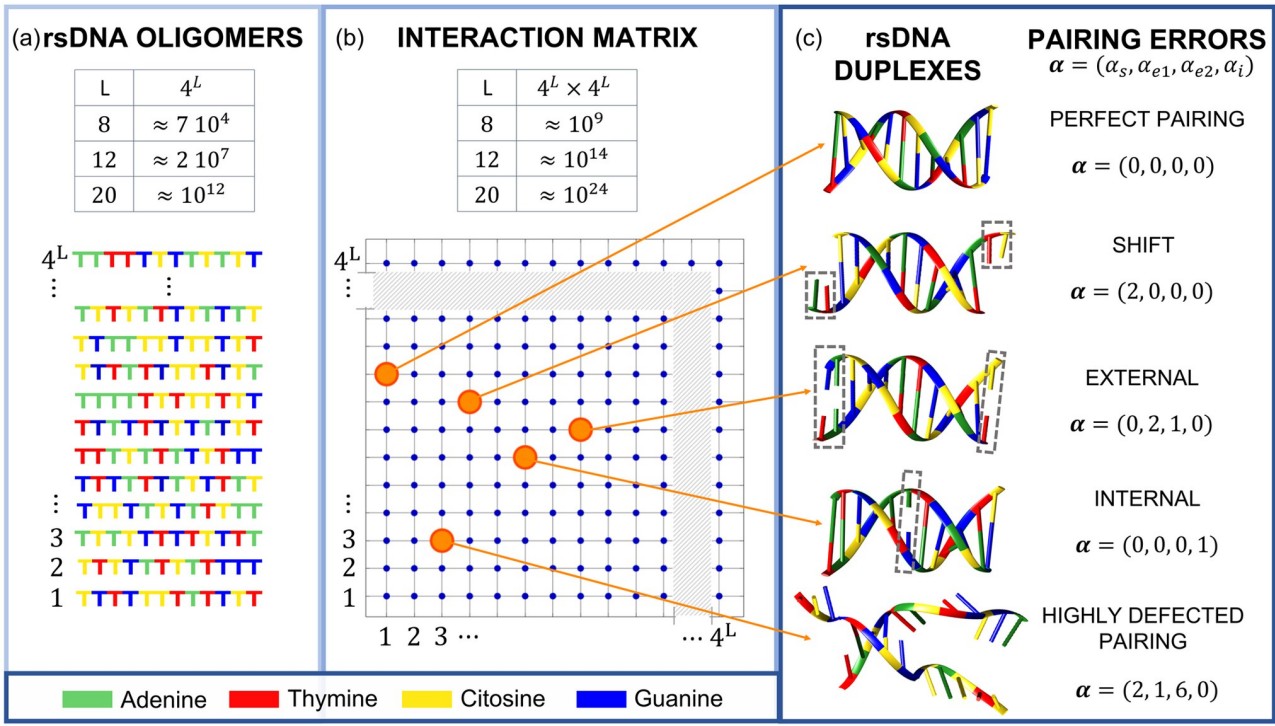

**Fig 1. Description of the system.** (a): Solutions of random-sequence DNA (rsDNA) oligomers of length L are mixtures made of $4^L$ distinct molecules, obtained by all the combinations of the four nucleobases, which are present at any position in the sequence with equal probability. (b): Each rsDNA oligomer can interact with $4^L$ different rsDNA oligomers, leading to a $4^L \times 4^L$ interaction matrix. Each dot in the matrix represents the most energetically favorable pairing between the two selected rsDNA oligomers, among all the possible mutual shifts. (c) Each position in the interaction matrix corresponds to a specific duplex motif, characterized by pairing errors which are here described by the parameters in $\boldsymbol{\alpha}$. The most probable duplex in the matrix is highly defected, as the last example in the panel.

rsDNA molecules is a quite unstable pair, with a small number of consequent paired bases (3 in case of L = 12, as in the bottom sketch of Fig 1c). However, even if less probable, more stable structures are formed, ranging from full complementarity, a condition that yields defectless helical duplexes (Fig 1c, top sketch), to duplexes with pairing mismatches of various types, listed in Fig 1c. To each duplex, with any pattern of defects, corresponds a binding free energy that can be computed from the sequences involved with standard tools. At the opposite end of the spectrum, there are rsDNA strands with no complementarity at all, as between an oligo made of only Ts and one made of only Cs.

In rsDNA at fixed L, the variety of binding possibilities increases with the number of allowed mismatches, while the binding energy decreases. As shown here, these two factors nearly compensate, leading to non-trivial competition between interaction strength and degeneracy. In a previous study of rsDNA [13], it was observed that, when $L > 12$, rsDNA solutions self-organize into liquid crystal phases. Given the mechanism by which these phases are formed in solutions of DNA oligomers [14, 15], this finding suggests that hybridization of rsDNA leads to duplexes with fairly well-paired terminals. This feature of rsDNA solutions remained speculative, with no experimental or statistical support.

Besides its value as a platform for exploring hybridization in crowded nucleic acid environments and for describing the network of interactions in superdiverse mixtures, the study of rsDNA is also relevant to the evaluation of scenarios for the origin of life. Indeed, if the RNA world hypothesis is correct, such a state had to be anticipated by a condition in which RNA

oligomers were abiotically synthesized with a large degree of randomness, from which ribo-zyme sequences could have been subsequently selected [16, 17]. Whether such molecular mix-tures could form WC pairs or whether hybridization was instead prevented by the large variety of species is an information that can shape the RNA world model itself, clarifying the role of complementarity and duplex formation in the prebiotic environment. [18].

In this paper, we study rsDNA solutions by a combination of three complementary strate-gies: (i) measurement of the overall degree of hybridization by UV absorbance as a function of temperature (T); (ii) measurements of the degree of hybridization and thermal stability of pairs of mutually complementary sequences mixed with rsDNA by fluorescence Contact-Quenching (CQ); (iii) development of a theoretical framework, based on a re-parametrization of the NN thermodynamic parameters, enabling quantitative predictions on rsDNA hybridiza-tion and pairing error statistics.

## Materials and methods

### Description of the system

8N, 12N and 20N rsDNA oligomers were synthesized on solid phase by an Äkta Oligopilot. The products were purified via dialysis against a $25mM$ NaCl solution and lyophilized. The samples were characterized by HPLC (see S2 Text). An analogous previous synthesis was char-acterized by MALDI-TOF mass spectroscopy [13]. Stock aqueous solutions were prepared at $c_{rsDNA} \approx 50g/l$, from which the final samples concentrations were obtained by dilution: ranging from $c_{rsDNA} = 0.04g/l$ to $c_{rsDNA} = 25g/l$ and with ionic strengths of $c_{NaCl} = 0.15M$, $c_{NaCl} = 0.45M$ and $c_{NaCl} = 1.0M$.

The statistics of the pairing quality have been explored by mixing rsDNA with a tagged pair of complementary DNA strands. Specifically, we have used the 8-base-long couple $8A^*$ and $8B^*$ and the 12-base-long couple $12A^*$ and $12B^*$, modified by 5'-Texas Red ($A^*$ oligomers) and 3'-6-FAM (Fluorescein) moieties ($B^*$ oligomers), respectively, so that upon hybridization the two fluorophores come in contact (see Fig G in S2 Text). Specific sequences are as follows. $8A^*$: TexasRed-5'-ACAGTCCT-3'. $8B^*$: 5'-AGGACTGT-3'-FAM. $12A^*$: TexasRed-5'-ACGA CAGTCCTG-3'. $12B^*$: 5'-CAGGACTGTCGT-3'-FAM. $8A^*$, $8B^*$, $12A^*$ and $12B^*$ were pur-chased from IDT.

### Using UV hyperchromicity to detect ensemble rsDNA melting

The overall degree of hybridization in rsDNA was evaluated by measuring the absorbance $A$ at the wavelength $\lambda = 258nm$. $A$ is obtained by averaging over an interval $\Delta\lambda = 3nm$. Experiments were performed with the Evolution 300 UV-Vis spectrophotometer from Thermo Scientific customized with a Quantum Northwest peltier hot/cold stage with hold temperature accuracy of $\pm0.05°C$. Experiments have been performed with $1°C/min$ heating and cooling rate. The cell holder was capable of hosting two different types of cell: standard quartz cuvettes with optical path length $\ell = 1cm$, adequate to investigate DNA solutions with $c \approx 0.02 - 0.04g/l$; microfluidic cells with $\ell = 10\mu m$ from Starna Scientific Ltd to investigate low volumes ($< 50\mu l$) of more concentrated DNA solutions ($c \approx 20 - 35g/l$) (see S2 Text). Absorbance data as func-tion of temperature, $A(T)$, were treated according to standard protocols [19] (see S2 Text) to extract the "ensemble" melting curve $\theta_e(T)$, i.e. the fraction of rsDNA oligomers forming duplexes (of any quality) at temperature $T$. The melting temperature $T_m$ is defined by $\theta_e(T_m) = 1/2$.

### Contact-quenching detects the pairing of specific sequences

Fluorescence-based measurements were used to detect how frequently a specific sequence is able to find its exact complementary strand in the midst of the rsDNA solution. This was done by mixing $8A^*$ and $8B^*$, in equal amount in a 8N solution, and similarly with $12A^*$ and $12B^*$ in 12N. Although the pair of fluorophores Texas Red and FAM were originally chosen to obtain FRET signal, we found that the dominant effect signaling their interaction is the so called "Contact Quenching" (CQ), *i.e.* the drop in fluorescence quantum yield of both fluorophores when the two fluorophores are in close proximity [20] (see S2 Text). In the case considered here, the quenching is deep (about 80% reduction for Texas Red and 50% for FAM) and can be easily exploited to extract the fraction $\theta_{AB}$ of $A^*$ oligomers that forms a defectless duplex with its complementary partner $B^*$.

To extract $\theta_{AB}(T)$ we monitored the quenching of the Texas Red emission, which is known to have a small T dependence [21]. Fluorescence emission vs. T was measured using the Applied Biosystems QuantStudio 5, a Real-Time PCR Instrument by Thermo Fisher Scientific. Calibration and normalization procedures to extract $\theta_{AB}$ from raw data are provided in S2 Text. They also include a thermodynamic characterization of the $8A^*$-$8B^*$ and $12A^*$-$12B^*$ duplexes, since their binding free energy $\Delta G_{A^*B^*}$ is slightly modified with respect to their untagged analogs because of the stabilizing effect of the fluorophores at the terminals. CQ experiments were performed by mixing fixed concentrations of both $A^*$ and $B^*$, $c_{fluo} = 100nM$, with rsDNA solutions prepared at $0.04g/l < c_{rsDNA} < 25g/l$, and ionic strengths $c_{NaCl} = 0.15M$ and $c_{NaCl} = 1.0M$. Measurements were performed in $15mM$ TRIS HCl at a pH of 7.4, to minimize fluorescence drift due to pH sensitivity of FAM [21].

In rsDNA solutions, each specific sequence is present at a concentration $c_{rsDNA}/4^L$. The stoichiometric ratio of each added fluorescent sequence ($A^*$ or $B^*$) and the same non-fluorescent sequence already present in the solution of rsDNA is:

$$\phi = \frac{c_{fluo}}{c_{rsDNA}/4^L}. \tag{1}$$

When $\phi = 1$, i.e. the amount of fluorescently labeled sequence equals that of the same sequence without tag, $\theta_{AB}$ offer an approximate evaluation of the degree in which errorless pairing is present within rsDNA solutions. At the same time, the measurement of the $\phi$ dependence of $\theta_{AB}(\phi)$ enables a detailed comparison with theoretical predictions.

## Experimental results

### Ensemble melting of rsDNA

Hyperchromicity in ultraviolet enables accessing the overall degree of hybridization in rsDNA solutions. Fig 2, green symbols, shows $\theta_e(T)$ for 12N, $c_{rsDNA} = 0.04g/l$, $c_{NaCl} = 1M$, which we compare, as reference, with the melting curves of binary mixtures of complementary DNA 12mers computed, with standard approaches, at two concentrations: $c_{DNA} = 0.02g/l$ (blue dashed line) and $c_{DNA} = 0.04/4^{12}\ g/l$ (red dashed line), both at $c_{NaCl} = 1M$. As visible, $\theta_e$ exhibits a behavior intermediate between the two. rsDNA duplexes are way more unstable (of $\approx 30°C$) than duplexes of complementary strands at equal total concentration, a clear manifestation of the selectivity of rsDNA pairing. At the same time, rsDNA duplexes appears about $5°C$ more thermally stable than the 12mer complementary duplexes when solubilized at the same concentration at which they are present in the rsDNA solution, an indication that the formation of defected pairing is a relevant feature of the hybridization of rsDNA, as also suggested by the milder slope of $\theta_e(T)$.

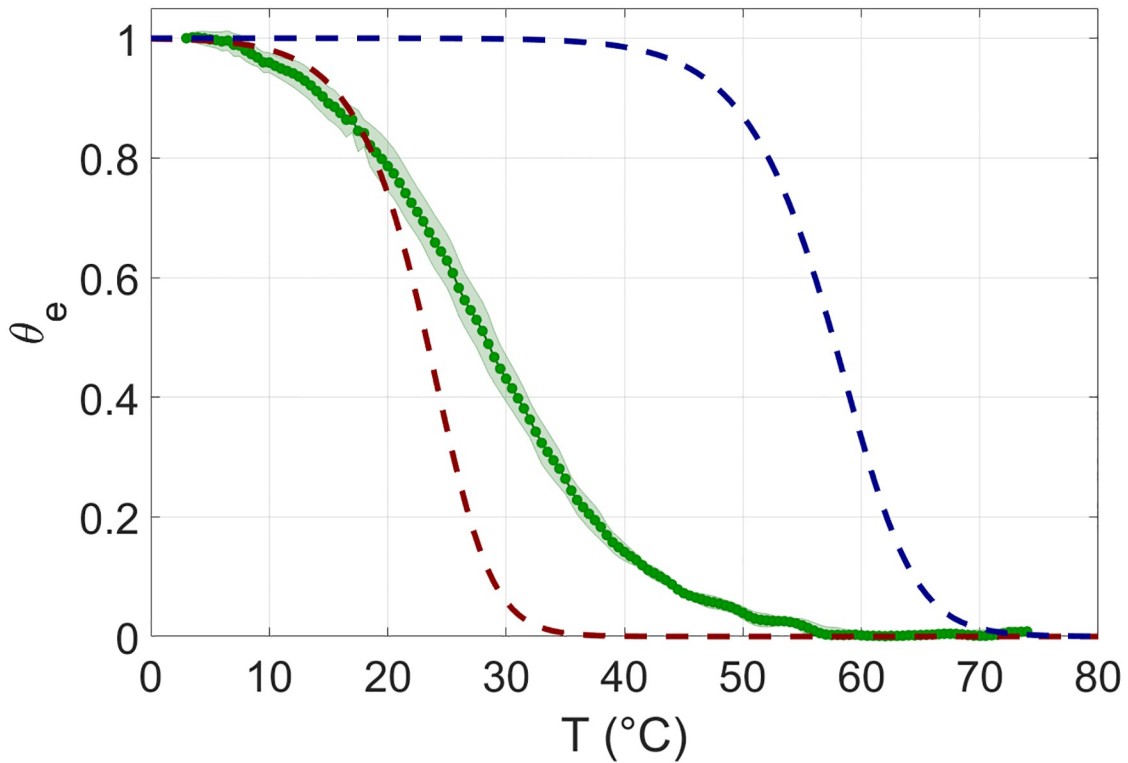

**Fig 2. Double strand vs. random sequence DNA melting.** Ensemble melting curve as a function of temperature. Green dots: measured ensemble melting of 12N at $c_{rsDNA}$ = 0.04 g/l. Shading marks experimental uncertainty resulting from the average over 8 experimental replicas. Dashed lines: theoretical melting predicted for equimolar solutions of two complementary 12mers at $c_{DNA}$ = 0.02 g/l each (dashed blue line) and $c_{DNA} = 0.04/4^{12} \approx 2.4 \ 10^{-9}$ g/l each (dashed red line). 12N $\theta_e(T)$ exhibits a behavior intermediate between the two. Dashed lines are obtained by averaging many melting curves of complementary 12mers. $c_{NaCl}$ = 1M in all curves.

In Fig 3a, $\theta_e(T)$ measured for 12N and 20N at $c_{rsDNA} \approx 0.04 g/l$ are shown for three different salt concentrations $c_{NaCl}$. Since $T_m$ decreases with $L$ but increases with $c_{rsDNA}$, to obtain reliable $\theta_e(T)$ for 8N, we performed melting experiments at a larger concentration, $c_{rsDNA} \approx 25g/l$, shown in Fig 3b. We find in all conditions $\theta_e(T)$ to depend on $T$ more mildly than in typical melting curves in binary solutions of complementary strands. Fig 3 also shows that $T_m$ of rsDNA grows with $c_{NaCl}$, with $L$ and with $c_{rsDNA}$, as it appears by comparing panels (a) and (b), in agreement with DNA melting in less complex systems [8, 22].

Experimental data shown here were taken at equilibrium. Attaining this condition is not trivial, since the lifetime of DNA duplexes dramatically depends on the length of the oligomers and on the concentration of the solutions [23, 24]. To approach equilibrium of 20N in dilute conditions, we considered only $\theta_e(T)$ measured upon heating after a long equilibration time at low T. Similar attention had to be given to the behavior of concentrated 12N solutions, as mentioned in the next Section. Further information on equilibrium conditions is provided in the S1 Text. The non-equilibrium behavior of rsDNA will be the topic of a future work.

## Probability of defectless duplexes

The study of $\theta_e(T)$ hints at hybridization in rsDNA as a combination of selectivity with a certain degree of defects in the duplex formation. However, $\theta_e(T)$ does not offer much insight on how probable it is to find duplexes with a certain pairing quality.Aiming at this kind of

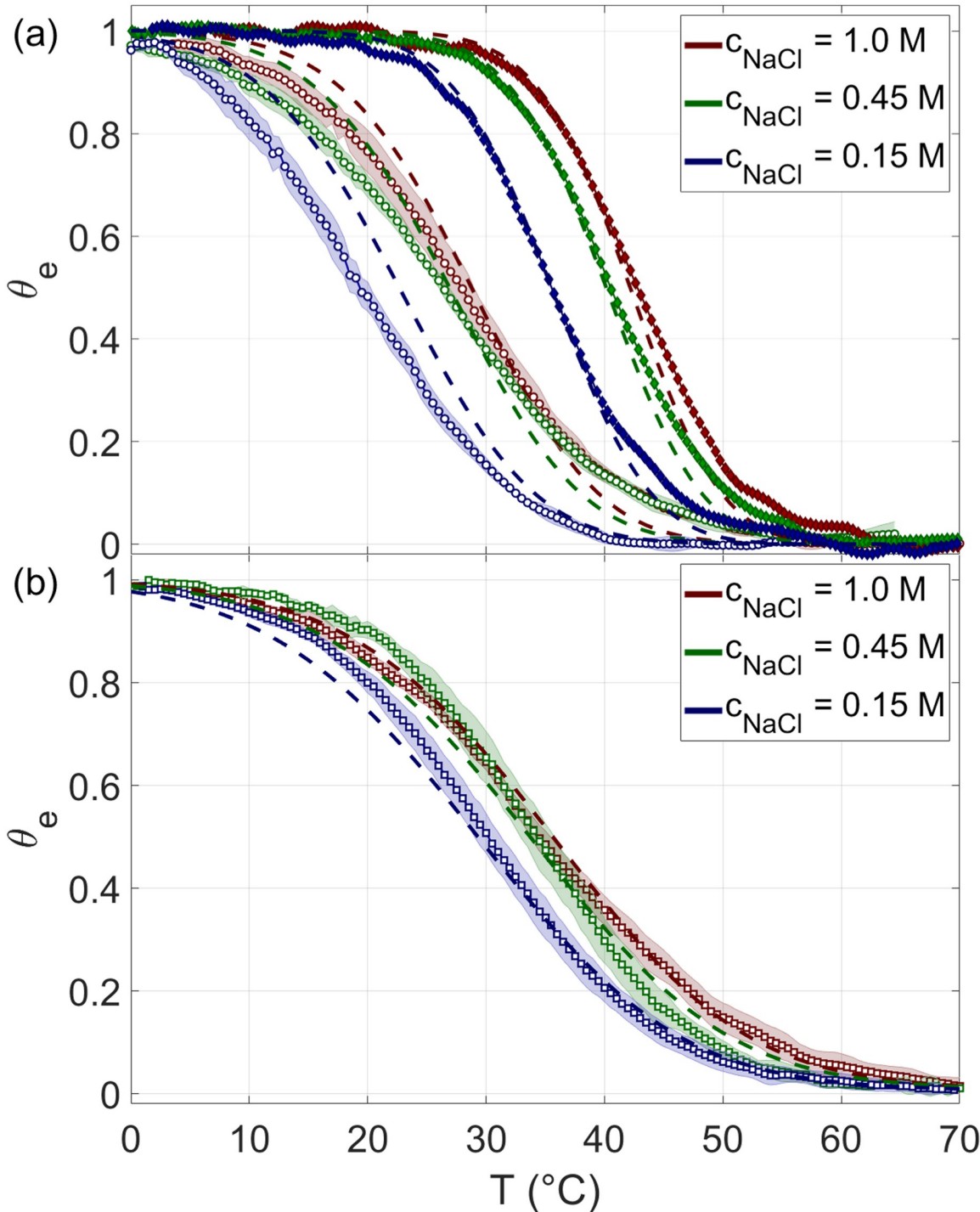

**Fig 3. Ensemble melting of rsDNA.** Measured ensemble melting curves of rsDNA at $c_{rsDNA}$ = 0.04 g/L for 12N (panel a, open circles), 20N (panel a, full diamonds) and 8N at $c_{rsDNA}$ = 25 g/L (panel b, open squares), at various salt concentrations: $c_{NaCl}$ = 0.15M (blue), $c_{NaCl}$ = 0.45M (green) and $c_{NaCl}$ = 1M (red). Shading marks experimental uncertainty resulting from the average over 6–8 experimental replicas for 8N and 12N, whereas, for 20N, just one experiment is shown as described in the text. Dashed lines, with same color code, are the theoretical predictions of Eq (7).

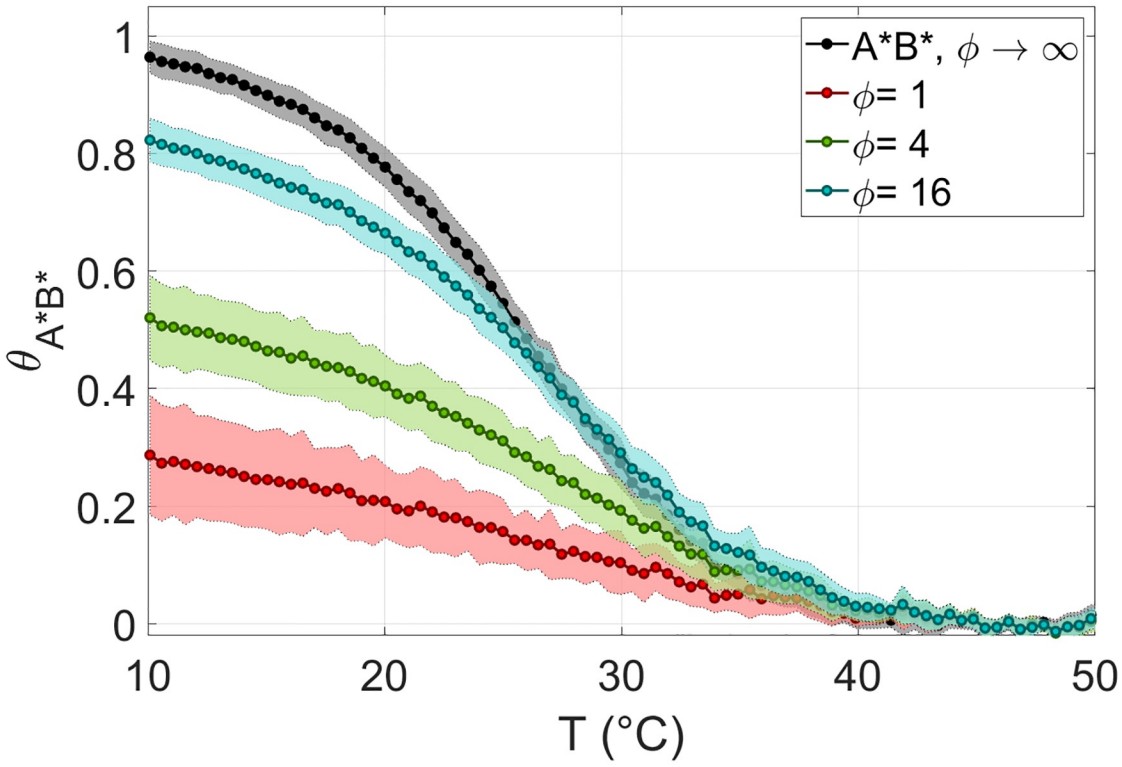

**Fig 4. Probability of defectless duplexes in rsDNA.** $\theta_{A^*B^*}$ Fraction of paired $8A^*8B^*$ in 8N measured with CQ experiments, at $c_{NaCl}$ = 0.15$M$. Colors correspond to different values of the stoichiometric ratio $\phi$ between the probes $A^*$ / $B^*$ and rsDNA strands. Black data is the melting curve of a neat $A^*B^*$ solution (without rsDNA). In all experiments $c_{fluo}$ = 100$nM$. Each data point is obtained as an average over 5–10 replications of the experiment. The corresponding standard deviation is reported as shaded regions.

information, CQ experiments in solutions of 8N and 12N were performed. We did not perform analogous measurement for the 20N both because of the artifacts due to non-equilibrium pairing and because of the small accessible range of $\phi$ ($4^{20}$ is a large number!).

Fig 4 shows the fraction of hybridized $8A^*$ and $8B^*$ in 8N, $\theta_{A^*B^*}(T)$, for various $\phi$. In the case of 8N we could reach $\phi = 1$, corresponding to $c_{rsDNA}$ = 16$g/l$. At $\phi = 1$ (red dots) the fraction of paired $A^*B^*$ reaches about 30% at low $T$. As $\phi$ increases, the fraction of $A^*B^*$ duplexes increases, as expected. The dependence of $\theta_{A^*B^*}(T)$ on $\phi$ is a useful tool to test our theoretical model, as discussed below.

Similar experiments, shown in S1 Text, have been carried on for $12A^*$ and $12B^*$ in 12N, where however in some conditions (largest $c_{rsDNA}$) we cannot reach equilibrium.

## Theoretical framework

Although the combination of the measured $\theta_e(T)$ and $\theta_{A^*B^*}(T)$ offers important insight on the quality of pairing within the rsDNA system, a deeper understanding of the driving mechanisms in this superdiverse environment requires a statistical model able to take into account the balance between binding energy and degeneracy. Indeed, the strongest binding energy is achieved in defectless duplexes, which are only formed with a fraction $1/4^L$ of the total number of sequences. On the contrary, defected duplexes are more weakly bound, but they can be assembled with a larger variety of sequences, i.e. the weaker the binding energy, generally, the larger its degeneracy.

The hybridization in simple systems formed by a limited number of sequences is well described by the current thermodynamic approaches, such as the NN model. Despite their accuracy, there is yet no theoretical frame to apply this knowledge to systems with high complexity such as the rsDNA, where more than $4^L \times 4^L$ interactions are involved. Explicitly computing the free energy for all pairs involved would be too computationally expensive. We thus develop a mean-field-type theoretical approach that uses averages of the thermodynamic parameters of the NN model and their re-parametrization on a simple counting of defects. With this approach we can compute $\theta_e(T)$ and $\theta_{CQ}(T)$ at equilibrium with no free parameters. Hence, a direct comparison between theory and experiments is achieved.

## Ensemble melting of rsDNA

We consider a rsDNA mixture containing $N$ chains in a volume $V$ (with a total concentration $c = N/V$). We assume the mixture to be perfectly balanced (see S2 Text), that is, eachsequence is present through $N/4^L$ copies. We also assume on-off hybridization, with nointermediate state between unbound and paired, which is justified given the limited length ($L \leq 20$) of the oligomers here considered [11]. Thus, a given couple ($i$ and $j$) interact with a set of $2L - 1$ distinct binding free energies $\Delta G_{ij}^{(\alpha_s)}$, depending on their mutual alignment: we use the shift parameter $\alpha_s$ with $-(L - 1) \leq \alpha_s \leq L - 1$ to express such alignment. Specifically, $\alpha_s = 0$ stands for the condition of perfect alignment, *i.e.* the 3' terminal base of strand $i$ is aligned with the 5' terminal base of strand $j$ and vice versa. Positive or negative $\alpha_s$ indicate an overhang on the 3' or 5' terminal, respectively.

We define the Boltzmann factor

$$\zeta_{ij}^{(\alpha_s)} \equiv [c] \, \exp(-\beta \Delta G_{ij}^{(\alpha_s)}), \tag{2}$$

where the brackets around the concentration denotes that it is measured in mol/L ($[c] = c/(\text{mol/L})$), and $\beta = (k_B T)^{-1}$ with $k_B$ being the Boltzmann constant. In real mixtures, some duplexes are very unlikely, with large $\Delta G$ and thus small $\zeta$.

In this setting, we can formally define, using the canonical distribution, the probability of any given hybridization state and the related partition function, which contains all the relevant statistical information of the system (see S3 Text). However, the partition function of the rsDNA system in the thermodynamic limit ($N \to \infty$) is not in a closed form. Consequently, an exact derivation of the fraction of paired oligomers $\theta_e$ in such a limit is not simple to obtain. This is thoroughly discussed in the S3 Text, where two opposite limits (high and low temperatures) are carried out allowing the derivation of analytical expressions to bypass this obstacle. Furthermore, an *Unification Ansatz* that matches the two approximations in their range of validity has been worked out. This theoretical approach allows us to compute $\theta_e^{(i)}(T)$, the approximated melting curve for the oligomers with sequence $i$ in the midst of all the $4^L$ species in the rsDNA solution (see S3 Text for details on the derivation).

$$\theta_e^{(i)} = 1 - \frac{2}{1 + \sqrt{1 + \frac{4}{4^L} \sum_{j=1}^{4^L} \sum_{\alpha_s=-(L-1)}^{L-1} \zeta_{ij}^{(\alpha_s)}}}. \tag{3}$$

This analytical expression has the same structure of the melting curve as a solution of self-complementary sequences (Eq 18a in Ref. [25]), with two relevant differences: the factor $1/4^L$ normalizing the concentration (the concentration of any specific sequence is $c/4^L$), and the double summation of the pairing weight of $i$: for all possible partner sequence $j$, and for all the possible shifts.

Since the explicit computation of all $(2L - 1) \times 4^L \times 4^L$ binding energies is prohibitive, and since our aim is to provide a statistical insight on the qualities of the double helices, we introduce a parametrization of the duplex quality, defining the pairing errors vector

$$\boldsymbol{\alpha} \equiv (\alpha_s, \alpha_{e1}, \alpha_{e2}, \alpha_i),$$ (4)

which describes the number and type of pairing errors in the rsDNA duplexes. Besides the shift parameter $\alpha_s$, $\boldsymbol{\alpha}$ includes the number of consecutive base mismatches at the two duplex terminals ($\alpha_{e1}$, $\alpha_{e2}$) and the number $\alpha_i$ of mismatches inside the duplex, as sketched in Fig 1c. This parametrization is useful to compute the degeneracy $g(L, \boldsymbol{\alpha})$, i.e., the number of sequences of length $L$ that can form a duplex characterized by $\boldsymbol{\alpha}$ with a given reference sequence:

$$g(L, \boldsymbol{\alpha}) = 4^{|\alpha_s|} 3^{(\alpha_{e1} + \alpha_{e2} + \alpha_i)} \binom{L - 2 - |\alpha_s| - \alpha_{e1} - \alpha_{e2}}{\alpha_i}.$$ (5)

Let us note that the exponential factors with base 3 and 4 indicate the number of different nucleobases that can occupy a site in the dangling end of the shift or lead to an internal or external mismatch, respectively, whereas the binomial coefficient accounts for all possible dispositions of the $\alpha_i$ mismatches in the internal region of the duplex.

The same set of parameters forming $\boldsymbol{\alpha}$ is used to evaluate the free energy of the pairings. The hybridization free energy is commonly computed on the base of the "Nearest Neighbor" approach [26], by which the binding enthalpy and entropy are obtained as a summation of quartets of neighboring nucleobases, their values depending on the specific bases that form it [8]. We propose a parametrization of the NN thermodynamics obtained by averaging over the quartets that allows us to compute $\Delta G^{(\boldsymbol{\alpha})}_{f_{CG}}$, an approximated value of $\Delta G^{(\alpha_s)}_{ij}$ based, not on detailed knowledge of the involved sequences but, just on $\boldsymbol{\alpha}$ and the fraction $f_{CG}$ of C or G bases in the sequence $i$. The reader can find in see S3 Text the computation of the average energetic parameters from the literature ones [8] and the salt correction [22]. An estimation of the error introduced in this simplification is shown in Fig 5, in which we show the melting curve of a solution of two 8mers with perfect complementarity. Therein, the melting curve computed using $\Delta G^{(\boldsymbol{\alpha})}_{f_{CG}}$ with $f_{CG} = 0.5$ and $\boldsymbol{\alpha} = \mathbf{0}$ (green line) is compared to the family of melting curves, computed with the traditional NN protocol, corresponding to a set of specific sequences with $f_{CG} = 0.5$ (dashed pink line and shadow). Our energetic description approximates the average melting curve with standard SantaLucia protocol with low discrepancy $\Delta T < 1°C$.

By inserting these parametrizations in Eq (3), the melting curve $\theta^{(i)}_e$ can be generalized to the melting curve of all the sequences in the rsDNA with a certain CG content:

$$\theta^{(f_{CG})}_e = 1 - \frac{2}{1 + \sqrt{1 + \frac{4}{4^L} \sum_{\boldsymbol{\alpha}} g(L, \boldsymbol{\alpha}) \zeta^{(\boldsymbol{\alpha})}_{f_{CG}}}},$$ (6)

where the product $g(L, \boldsymbol{\alpha}) \zeta^{(\boldsymbol{\alpha})}_{f_{CG}}$ expresses the statistical weight of the pairings of a sequence with CG presence of $f_{CG}$ with all the sequences leading to the formation of a duplex with quality $\boldsymbol{\alpha}$. Note that $\Sigma_{\boldsymbol{\alpha}}$ refers to summation over all the possible pairs yielding $\boldsymbol{\alpha}$. Fig 5 shows the set of $\theta^{(f_{CG})}_e$ with $f_{CG}$ ranging from 0 to 1 in the case of 8N.

The ensemble melting is obtained by the average of $\theta_e^{(f_{CG})}$ over the possible values of $f_{CG}$, weighted with the probability $p_L(f_{CG})$ of having a sequence with $f_{CG}$ in the rsDNA:

$$\theta_e = \sum_{f_{CG}} p_L(f_{CG})\, \theta_e^{f_{(CG)}}$$

$$= 1 - \frac{1}{2^L} \sum_{f_{CG}} \binom{L}{f_{CG} \cdot L}$$

$$\times \frac{2}{1 + \sqrt{1 + \frac{4}{4^L} \sum_{\boldsymbol{\alpha}} g(L, \boldsymbol{\alpha}) \zeta_{f_{CG}}^{(\boldsymbol{\alpha})}}} \,. \tag{7}$$

The summation is over all possible fractions, that is, the product $f_{CG} \cdot L$ sweeps the integer numbers from 0 to L. Fig 5 shows $\theta_e$ (red line) for 8N. Two features are clearly noticeable: the ensemble $T_m$ predicted for rsDNA is much lower than those of complementary strands at the same concentration and ionic strength (dashed line), in agreement with experimental observations; the $T$ dependence of $\theta_e$ is milder, reflecting the variety of energies involved in the various duplexes that can be formed in rsDNA solutions. The choice of splitting the statistical summations in two steps enlightens the relevance of both averaging the free energy for fixed $f_{CG}$ and

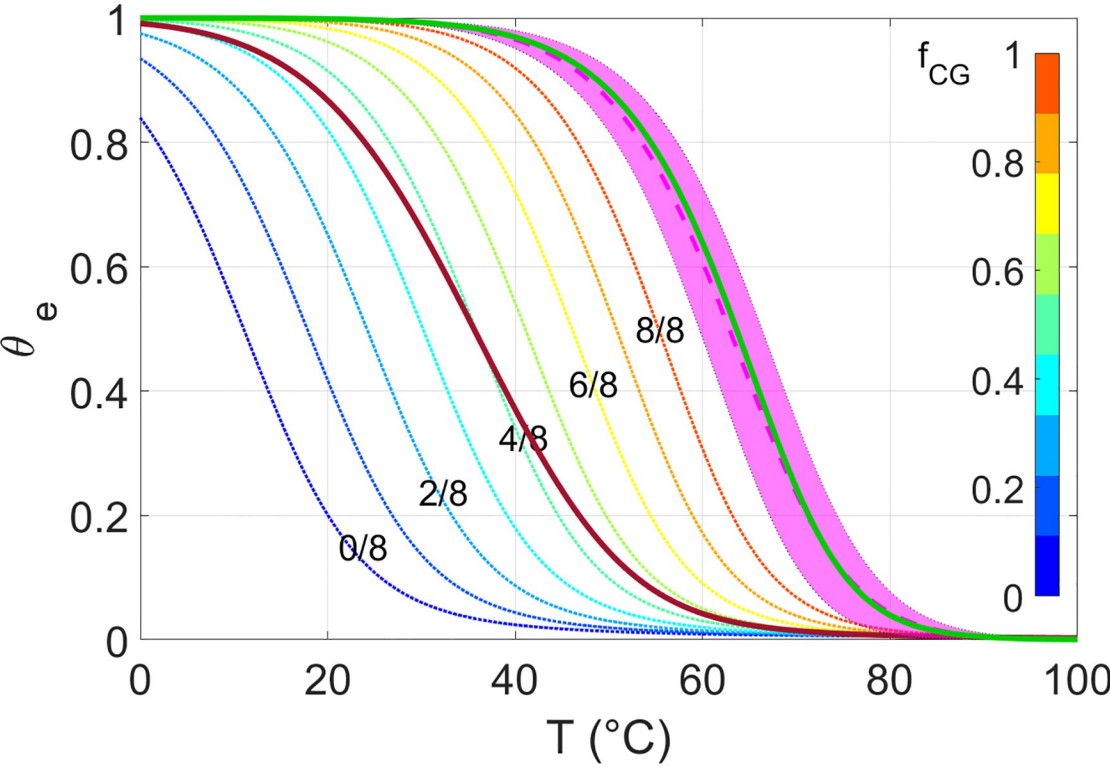

**Fig 5. Theoretical predictions of melting curves, computed for DNA 8mers at 25g/L and 1M NaCl.** The green solid line stands for the melting curve predicted for a pair of complementary strands when using the energy obtained from our parametrization, with $\boldsymbol{\alpha} = \mathbf{0}$ and $f_{CG} = 0.5$. The purple dashed-line and related shading are, respectively, the mean value and the standard deviation computed using the NN model over a set of 40 melting curves of distinct DNA sequences with fixed $f_{CG} = 0.5$. The dotted lines are the melting curve $\theta_e^{(f_{CG})}$ of 8N with different values of $f_{CG}$ (Eq (6)), as specified in the colorbar. The solid dark red line is the ensemble melting curve $\theta_e$, obtained by the average of $\theta_e^{(f_{CG})}$ over the possible values of $f_{CG}$ (as given by Eq (7)).

of averaging the melting curves over $f_{CG}$. While the latter is apparent upon inspecting Fig 5, further insight on the former can be obtained by Fig A in S3 Text, where the melting curve inclusive of all summations for $f_{CG}$ = 0.5 is compared with simplified approaches, which are found to differ both in $T_m$ and in the shape of the melting curve.

## Probability of defectless and defected duplexes

**Neat rsDNA solutions.** By using the parametrization introduced above, it is possible to evaluate the fraction $\theta_{\boldsymbol{\alpha}}^{(f_{CG})}$ of strands involved in duplexes having any specific type of pairing $\boldsymbol{\alpha}$, for sequences with $f_{CG}$. This is done by weighting the total fraction of paired oligomers $\theta_e^{(f_{CG})}$ with the statistical weight of the specific defect class, i.e.,

$$\theta_{\boldsymbol{\alpha}}^{(f_{CG})} = \frac{g(L, \boldsymbol{\alpha})\zeta_{f_{CG}}^{\zeta(\boldsymbol{\alpha})}}{\sum_{\boldsymbol{\alpha'}} g(L, \boldsymbol{\alpha'})\zeta_{f_{CG}}^{\zeta(\boldsymbol{\alpha'})}} \theta_e^{(f_{CG})}. \tag{8}$$

The fraction $\theta_{\boldsymbol{\alpha}}$ of duplexes having a given $\boldsymbol{\alpha}$ in the whole rsDNA solution is the averaged $\theta_{\boldsymbol{\alpha}}^{(f_{CG})}$, weighted with the probability $p_L(f_{CG})$,

$$\begin{aligned} \theta_{\boldsymbol{\alpha}} &= \sum_{f_{CG}} p_L(f_{CG}) \, \theta_{\boldsymbol{\alpha}}^{(f_{CG})} = \\ &= \frac{1}{2^L} \sum_{f_{CG}} \binom{L}{f_{CG} \cdot L} \cdot \frac{g(L, \boldsymbol{\alpha})\zeta_{f_{CG}}^{\zeta(\boldsymbol{\alpha})}}{\sum_{\boldsymbol{\alpha'}} g(L, \boldsymbol{\alpha'})\zeta_{f_{CG}}^{\zeta(\boldsymbol{\alpha'})}} \theta_e^{(f_{CG})} \end{aligned} \tag{9}$$

Having access to $\theta_{\boldsymbol{\alpha}}$, we can rank the defects based on their probability. This is shown in Fig 6a where we plot the six most frequent errors in 12N as a function of $T$. Perfect (defectless) duplexes are not dominant, but also not negligible, exceeding 10% at the lowest $T$ considered. The most frequent form of error is at the terminals as a result of their reduced energy cost compared to internal mismatches.

By suitable summation of $\theta_{\boldsymbol{\alpha}}$, it is possible to determine the fraction of duplexes having a total of defects $|\boldsymbol{\alpha}| \equiv |\alpha_s| + \alpha_{e1} + \alpha_{e2} + \alpha_i$. The resulting $\theta_{|\boldsymbol{\alpha}|}$ are shown in Fig 6b. We find the fraction of duplexes with $|\boldsymbol{\alpha}|$ = 1 to be dominant, while the fraction of duplexes with more than 2 errors becomes negligible at low $T$. It is also interesting to notice that $\theta_{|\boldsymbol{\alpha}|}$ computed for the three considered values of $L$ converges to the same value at low T. This phenomenon can be understood as a consequence of mainly two reasons. First, $g(L, \boldsymbol{\alpha})$ does not depend on $L$ when only dangling ends and external mismatches, the dominant form of pairing errors, are present (see Eq (5)). Second, internal mismatches are instead nearly negligible at low temperatures in this range of $L$ (Fig 6a). An analytic derivation of this low $T$ limit is reported in the S3 Text.

**Modeling the contact quenching experiments.** Fig 6 shows the fraction of duplexes with a certain quality $\boldsymbol{\alpha}$ among all the possible kinds of duplexes forming the rsDNA mixture. However, in CQ experiments, the rsDNA solution is enriched by the presence of the labelled strands $A^*$ and $B^*$ at a concentration expressed by the ratio $\phi$. The fraction of $A^*B^*$ pairs at a given $\phi$ can be predicted through an adequate extension of the model,

$$\theta_{A^*B^*} = \frac{\phi\zeta_{A^*B^*}}{\phi\zeta_{A^*B^*} + \sum_{\boldsymbol{\alpha}} g(L, \boldsymbol{\alpha})\zeta_{f_{CG}}^{\zeta(\boldsymbol{\alpha})}} \theta_e^{(CQ)}, \tag{10}$$

where $\zeta_{A^*B^*} = [c] \exp(-\beta\Delta G_{A^*B^*})$ is the Boltzmann weight of the $A^*B^*$ couple, $f_{CG}$ is the fraction of C or G bases in the $A^*B^*$ sequences and $\theta_e^{(CQ)}$ is the fraction of paired $A^*$ or $B^*$, either to each other or to any other oligomer of the rsDNA solution, that can be computed generalizing Eq

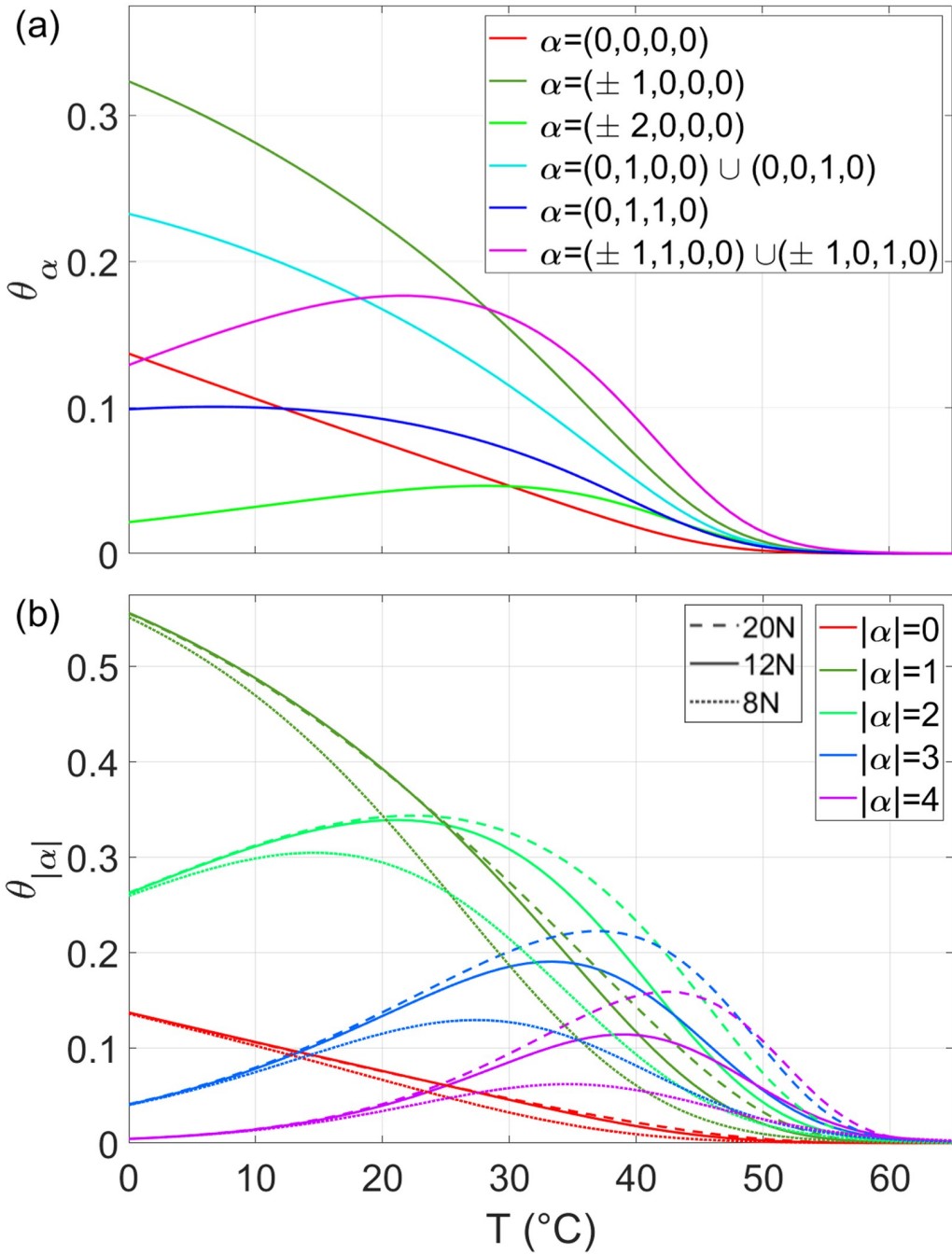

**Fig 6. Pairing statistics in rsDNA.** Theoretical predictions of the fraction of perfect and defected rsDNA duplexes (as given by Eq (9)), parametrized by $\alpha$. (a): the six most probable duplex motifs in 12N. (b): fraction of duplexes with a total number of unpaired bases $|\alpha|$, in 8N (dotted lines), 12N (continuous lines), 20N (dashed lines). $c_{NaCl} = 1M$ and $c_{rsDNA} = 25g/l$.

(6) for $\theta_e^{(f_{CG})}$,

$$\theta_e^{(CQ)} = 1 - \frac{2}{1 + \sqrt{1 + \frac{4}{4^L}\left[\phi\zeta_{A^*B^*} + \sum_{\boldsymbol{\alpha}} g(L, \boldsymbol{\alpha})\zeta_{f_{CG}}^{(\boldsymbol{\alpha})}\right]}}.$$ (11)

Further details on the pairing energies are given in the S2 Text. It can be easily seen that, in the limit of $\phi \to \infty$, $\theta_e^{(CQ)}$ correctly converges to the melting curve of a couple of complementary strands in equal concentration.

## Discussion

The model we developed here provides theoretical predictions directly comparable to experimental observations for both the ensemble melting and the formation of defectless duplexes in rsDNA solutions.In Fig 3, the theoretically computed $\theta_e(T)$ (dashed lines) are compared to the measured $\theta_e(T)$ (symbols). The agreement is very good, especially if we take into account that the model has no free parameters. Fig 7 compares the experimental melting temperatures, $T_m$, to its theoretical prediction as a function of the parameters here considered: $L$, $c_{rsDNA}$, $c_{NaCl}$. The average deviation between predicted and observed $T_m$ is within $1°C$, or within $0.5°C$ if we exclude the data point of 12N at $c_{NaCl} = 0.15M$, in which $T_m$ is the lowest and thus more difficult to determine because of the narrow $T$ interval available to define the low-$T$ baseline (see blue open circles in Fig 3a). The difference between computed and measured $T_m$ is comparable

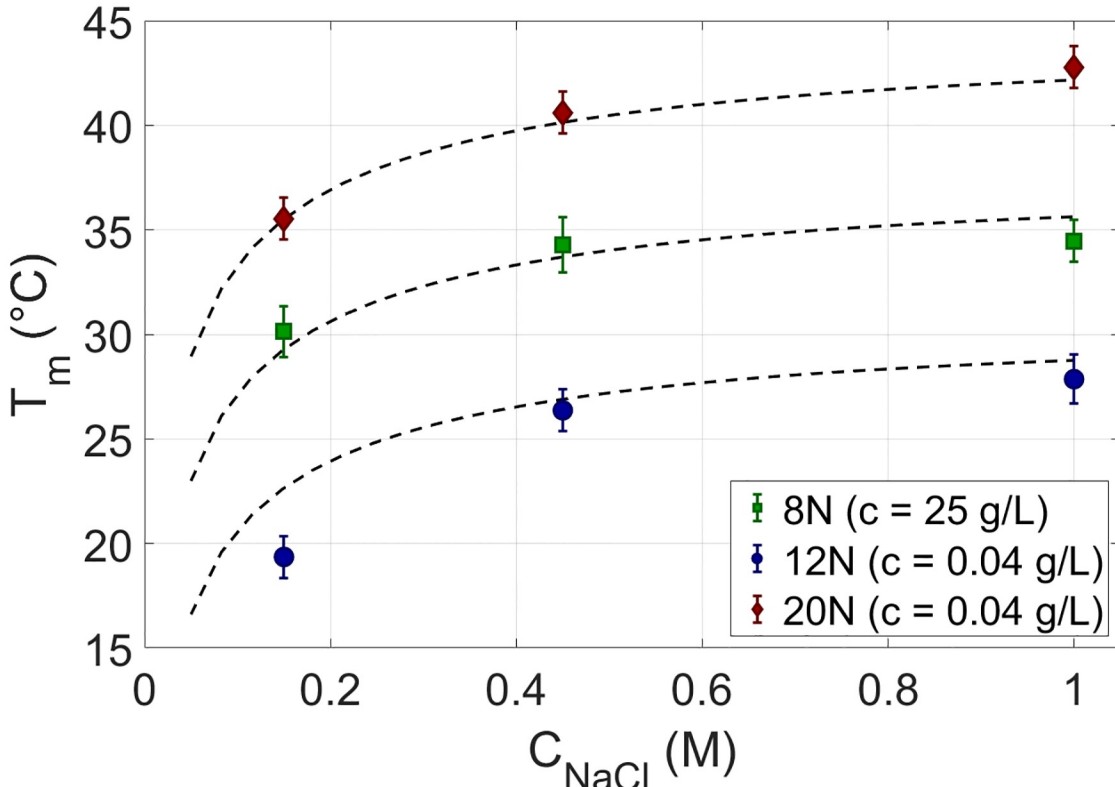

**Fig 7. Melting temperatures for rsDNA: Theory vs experiment.** Comparison of measured and predicted $T_m$ for rsDNA solutions. Symbols: experimental $T_m$ obtained from the melting curves of 8N, 12N and 20N (Fig 3), as function of salt, $c_{NaCl}$. Conditions are specified in the legend. Dashed lines: theoretical predictions of Eq (7).

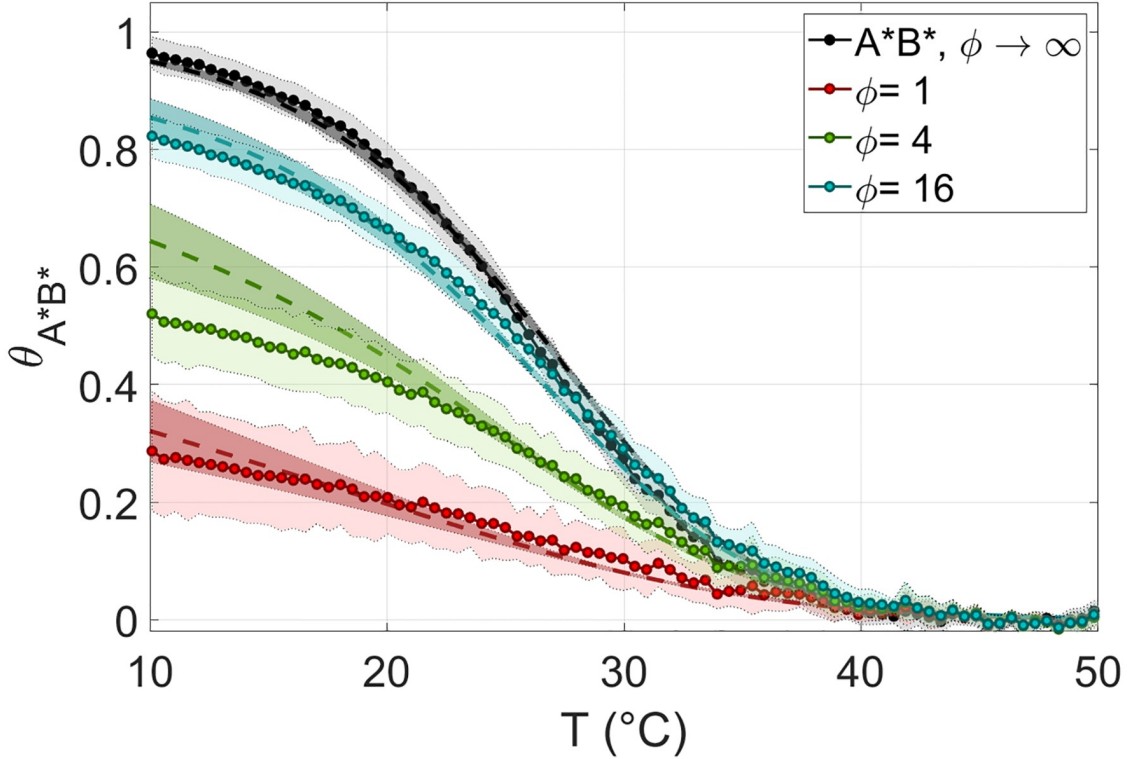

**Fig 8. Probability of defectless duplexes in rsDNA: Theory vs experiment.** Fraction of paired $8A^*8B^*$ in 8N, $\theta_{A^*B^*}$, as determined via CQ experiments (dots and light shading, as in Fig 4) and from the model (dashed lines and dark shading), at $c_{NaCl} = 0.15M$ and for several $\phi$ values (colors, see legend). Black data, lines and shadings: melting in $8A^*+8B^*$ solutions, in the absence of rsDNA. Shaded regions of the theoretical predictions are obtained from the experimental uncertainty on the pairing energy between $A^*$ and $B^{**}$ (see S2 Text).

with the typical errors in predicting $T_m$ for any given specific sequence with standard thermo-dynamic approach [22].

The key result of the model is the non-trivial $T$ dependence of the statistics for pairing qual-ity in rsDNA solutions, which is codified into $\theta_{\boldsymbol{\alpha}}(T)$, the fraction of duplexes with quality $\boldsymbol{\alpha}$, as given by Eq (9) and shown in Fig 6. $\theta_e(T)$ reflects the complexity of the interactions in such superdiverse environment, but only as an ensemble feature. A much more stringent test is provided by the comparison between CQ experiments, *i.e.*, the measured and predicted $\theta_{A^*B^*}(T, \phi)$. In Fig 8 we compare the CQ data already shown in Fig 4 for 8N at $c_{NaCl} = 0.15M$ with the predicted $\theta_{A^*B^*}(T)$ for several values of $\phi$. The agreement is good, compatible with the range of uncertainty of both experiment and model, the latter being a consequence of the uncertainty on the free energy associated to the $8A^*8B^*$ duplex formation.

In Fig 9 we show measured and predicted $\theta_{A^*B^*}$ as a function $\phi$ for 8N at $T \approx 15°C$ at two ionic strengths, $c_{NaCl} = 0.15M$ and $c_{NaCl} = 1M$. Noticeably, the ionic strength has little effect on $\theta_{A^*B^*}(T, \phi)$, much less than on $T_m$. This is because the salt contribution to the statistical weights is similar for the most probable pairings. Since at low $T$ the pairing probability is approximately the ratio between statistical weights, the salt contribution effectively cancels (see S3 Text).

The success of our model—which has no free parameter—in describing rsDNA solutions might be surprising, given the level of approximation introduced in the energy parametriza-tion on $\boldsymbol{\alpha}$. Indeed, the molecular diversity of rsDNA makes its behavior intrinsically averaged between all possible pairings patterns, rendering our parametrization, built on averages,

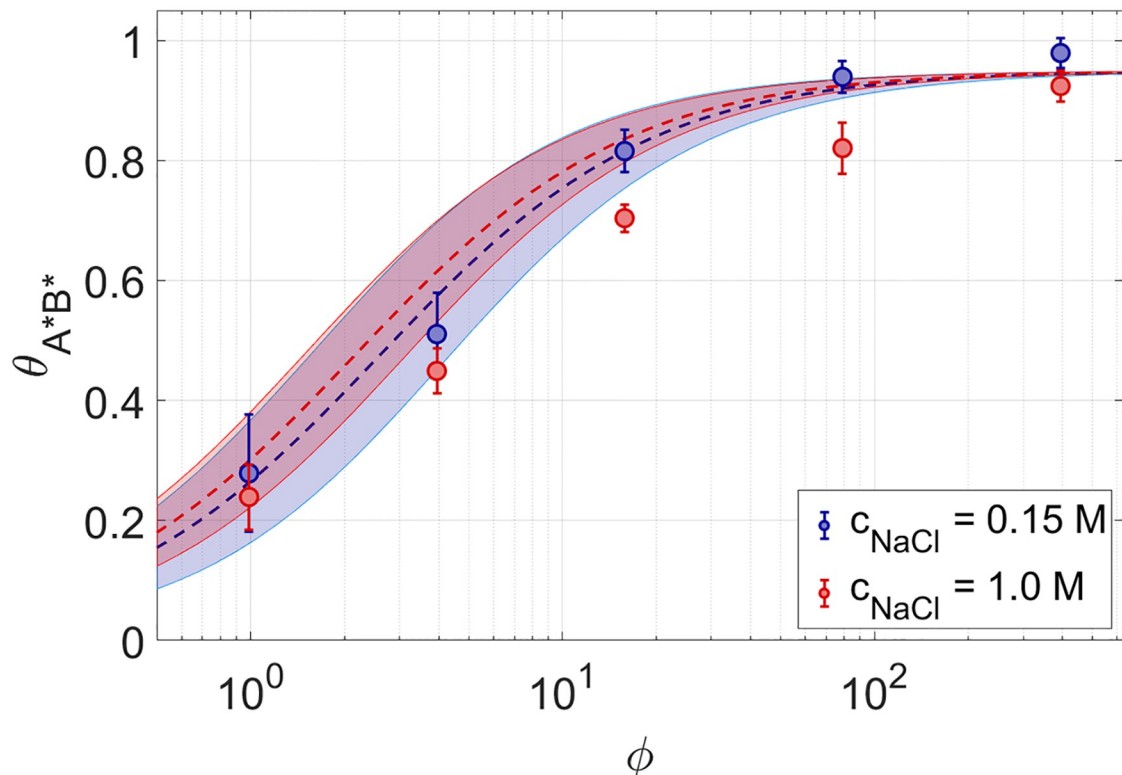

**Fig 9. Probability of defectless duplexes in rsDNA: Salt dependence.** Fraction of paired $8A^*8B^*$ as a function of $\phi$, expressing their dilution in 8N, at $T = 15°C$ for $c_{NaCl} = 0.15M$ (blue dots) and $c_{NaCl} = 1.0M$ (red dots). Dashed lines: theoretical predictions, with the shaded regions obtained from the experimental uncertainty on the pairing energy between A*B*, (see S2 Text).

particularly adequate. We would also like to point out that the model validity is limited to equilibrium conditions, as we documented above and in the S1 Text. Also, the model assumes unlimited molecular availability, and does not include the effects of competitive binding which could arise from constrained stoichiometric ratios in limited pools of molecules. The successful comparison with experiments indicates that rsDNA with short enough chain length satisfies all these requirements. In systems with longer molecules, out-of-equilibrium conditions and hybridization states with more than one helical region could become relevant [11].

The agreement with observations also validates the predicted pairing distributions $\theta_{\alpha}(T)$ and $\theta_{|\alpha|}(T)$ shown in Fig 6, which are worth discussing further. These distributions indicate that, in a superdiverse environment of random sequence oligonucleotides, the selectivity afforded by the free energy of base-pairing is "marginal". Specifically, the resulting pairing is good, but not perfect, with the majority of sequences being defected, but with less than two pairing errors. Defectless pairing involves at most a fraction of $\approx 14\%$ of the rsDNA strands.

The pairing statistics of rsDNA largely depends on the compensation between two opposing factors: (i) the degeneracy, $g(L, \boldsymbol{\alpha})$, which grows in a nearly exponential way with the total number of pairing errors (see Eq (5)), thus favoring the formation of defected duplexes; and (ii) the binding free energy, $\Delta G$, which increases approximately linearly with the number of defects (see S3 Text), yielding a Boltzmann factor with an exponential advantage to duplexes with less defects (see Eq (2)). In DNA duplexes formation, these two factors nearly balance, with a partial dominance of the energetic component, a condition leading to the smooth $\alpha$ and $T$ dependence of the probability distributions. The result would change if the degeneracy

grows faster or slower than an exponential with the binding energy. An example of this latter condition is given by the selectivity of PCR primers within the genome, in which the set of competing bindings is limited with respect to the random situation, thus enabling a strong dominance of defectless primer binding.

An obvious question is how critical is this marginal condition, and whether modifications in the nucleobase structure, and thus of pairing and stacking energy, could significantly improve selectivity. To answer this question we computed $\theta_{\alpha}(T)$ by assuming that all pairing energies equal that of CG (which is roughly double of that of AT). We find a significant, but not dramatic, increment in $\theta_0$, that at low $T$ reaches 0.3 (see Fig D in S1 Text), indicating that the basic features of $\theta_{\alpha}(T)$ are stable within the range of energies involved in natural and artificial nucleobase binding [7]. As a further test of the pairing efficiency in random systems, we computed $\theta_0(T)$ by considering random systems formed by only 2 bases, instead of the four natural ones, using energetic parameters intermediate between AT and CG (see Fig E in S1 Text). Even in this condition, where the degeneracy of the defected duplexes is strongly reduced, the fraction of perfect pairs is larger but still well under 0.4. When, on the contrary, the number of bases are increased (still assuming a WC-type pairing role), $\theta_0$ markedly decreases. These observations strengthen the notion that, in the range of pairing energies of nucleic acids, the presence of randomness appears to be the dominant factor in determining the quality of the pairs.

These observations sets a reference for the selectivity in contexts of strong randomness and heterogeneity of sequences such as those that have likely characterized the origin of life and RNA world. Whatever the mechanisms of chain amplification and lengthening, and whatever base pair variants were at the time available, they could not have relied on levels of selectivity much better than those reported here. We previously proposed that one of such mechanisms leading to the formation of long nucleic acid chains could exploit the symmetry breaking and formation of molecular column due to liquid crystal ordering [27, 28]. rsDNA can indeed form, in given conditions, columnar liquid crystals [13]. How the distribution of pairing quality here discussed can be compatible with liquid crystal formation appears as a subtle matter that will be the topic of a forthcoming work.

## Conclusion

We introduced rsDNA solutions as a model system of superdiverse mixture, enabling the study of interactions and pair formations in the midst of a huge amount of competing molecular species, a condition offering a conceptual paradigm for the molecular variety and selectivity of biological environments. In the analysis of rsDNA solutions, we could take advantage of the limited polymer heterogeneity given by the four nucleobases, of the rather simple and highly characterized pairing rule, of the availability of solid-state synthesis and of the variety of experimental tools.

This combination of factors enabled us to experimentally characterize, and theoretically describe, the selectivity of pairing, and found that the majority of rsDNA duplexes contain pairing errors, but limited to one or two per duplex, a condition that still grants a reliable stability to the structures.

Based on the success in the description of rsDNA, we applied our approach to the description of the selective pairing in the context of the PCR technology and miRNA based gene regulation, which is the topic of a forthcoming publication.

Extending this statistical approach to biomolecular superdiverse systems closer to cell environments, characterized by a less dramatic diversity but by more complex and less defined interactions, will be the challenging development of this work.

## Supporting information

**S1 Text. Further results.** Evidence of Out-of-Equilibrium Conditions; Perfect pairing probability with different energies and number of nucleobases types. **Fig A: Evidence of out-of-equilibrium behavior for 12N.** $T$ dependence of the fraction of duplexed strands $\theta_e$ measured while heating and cooling at $1°C/min$. **Fig B: Evidence of out-of-equilibrium behavior for 20N.** Absorbance vs. $T$ measured upon heating after one month equilibration at $4°C$. **Fig C: $\theta_{A^*B^*}$, fraction of paired $12A^* - 12B^*$ in 12N**, determined from the model and via CQ experiments with different cooling rates. **Fig D: $\theta_0^{(f_{CG})}$ computed with different values of $f_{CG}$ in 8N. Fig E: $\theta_0^{(n_b)}$ computed with different values of $n_b$ in 12N.**
(PDF)

**S2 Text. Materials and methods.** Characterization of rsDNA synthesis; Measurement of rsDNA concentration; UV Absorbance: Experimental Setup; Analysis of UV Absorbance Data; Characterization of $A^*$ and $B^*$ Fluorescence; Contact-Quenching Data Analysis; Free Energy of $A^*B^*$ Duplex. **Fig A: HPLC traces** of 12N compared with two different 12mers. HPLC traces of 8N, 12N and 20N. **Fig B: Quartz microfluidic. Quantum Northwest Peltier. Fig C: Temperature calibration of $T$** measured by a thermistor in contact with the microfluidic cell vs. the internal control $T_{peltier}$. **Fig D: Steps in the analysis of absorbance data $A(T)$ of 12N** used to extract the melting curves $\theta_e$. **Fig E: Fluorescence Emission Spectra of labeled DNA systems:** $A^*$, $B^*$, $A^*B^*$, $A^*B$ and $AB^*$. **Fig F: Absorbance Spectra of labeled DNA systems:** $A^*$, $B^*$, $A^*B^*$, $A^*B$ and $AB^*$. **Fig G: Simplified representation of the $12A^*B^*$ duplex**, showing the relative size of DNA duplex, linker and fluorescent moieties FAM and TexasRed. **Fig H: fluorescence intensity of TexasRed vs. $T$** in a solution of $12A^* + 12B^*$. **Fig I: Normalized fluorescence intensity of TexasRed** in a solution of $12A^*$, $12B^*$ and 12B. **Fig J: Average normalized fluorescence of $8A^*+8B^*$.** The fit enables determining the linear drift at low temperatures, corresponding to the signal of fully paired $A^*B^*$. **Fig K: $T_m$ vs $A^*B^*$ concentration measured by CQ for the following systems**.
(PDF)

**S3 Text. Comprehensive Description of the theoretical model.** Partition Function; Melting Curve Approximations for rsDNA solution; Energetic Parametrization based on $\boldsymbol{\alpha}$; Pairing Statistics with $\alpha_i = 0$; Effects of the Ionic Strength on the Pairing Statistics. **Fig A: Melting Curves of rsDNA**, according to the *Low T Approximation*, *High T Approximation* and *Unification Ansatz*.
(PDF)

## Acknowledgments

We thank N.A. Clark for useful discussions and E. Toffolo for her precious guidance in using PCR equipment for CQ measurements.

## Author Contributions

**Conceptualization:** Tommaso P. Fraccia, Amos Maritan, Samir Suweis, Tommaso Bellini.

**Formal analysis:** Stefano Marni, Carlos A. Plata, Amos Maritan, Samir Suweis.

**Funding acquisition:** Tommaso Bellini.

**Investigation:** Simone Di Leo, Stefano Marni, Carlos A. Plata, Tommaso Bellini.

**Methodology:** Simone Di Leo, Stefano Marni, Carlos A. Plata, Samir Suweis, Tommaso Bellini.

**Project administration:** Samir Suweis, Tommaso Bellini.

**Resources:** Gregory P. Smith, Tommaso Bellini.

**Software:** Stefano Marni, Carlos A. Plata.

**Supervision:** Tommaso Bellini.

**Validation:** Simone Di Leo, Stefano Marni, Carlos A. Plata, Tommaso P. Fraccia, Amos Maritan, Samir Suweis, Tommaso Bellini.

**Writing – original draft:** Simone Di Leo, Stefano Marni, Carlos A. Plata, Samir Suweis, Tommaso Bellini.

**Writing – review & editing:** Simone Di Leo, Stefano Marni, Carlos A. Plata, Tommaso P. Fraccia, Gregory P. Smith, Amos Maritan, Samir Suweis, Tommaso Bellini.

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
