## [Decision Letter · Decision Letter 0]

14 Feb 2022

Dear prof Bellini,

Thank you very much for submitting your manuscript "Pairing Statistics and Melting of Random DNA Oligomers: finding your Partner in Superdiverse Environments" for consideration at PLOS Computational Biology. As with all papers reviewed by the journal, your manuscript was reviewed by members of the editorial board and by several independent reviewers. The reviewers appreciated the attention to an important topic. Based on the reviews, we are likely to accept this manuscript for publication, providing that you modify the manuscript according to the review recommendations.

Sincerely,

Eugene I. Shakhnovich

Guest Editor

PLOS Computational Biology

Nir Ben-Tal

Deputy Editor

PLOS Computational Biology

[LINK]

Reviewer's Responses to Questions

**Comments to the Authors:**

Reviewer #1: The authors present a combined experimental and theoretical thermodynamic study of a solution consisting of all possible L-nucleotide-long DNA oligomers (L being 8, 12 and 20) generated by the random synthesis from a mixture of equal proportion of four nucleotide precursors. Authors’ main conclusion is that, despite the presence of a number of chains that can form mismatched duplexes with the given chain, which could prevent it from finding the complementary partner, the yield of fully matched duplexes at low enough temperatures is pretty high.

Major comments

1. The authors failed to convince this reviewer that a pretty artificial system they consider deserves so much attention. In Introduction, they claim that the system they consider is relevant to the pre-biological evolution consideration within the framework of the RNA World hypothesis. But this claim looks too far-fetched to me.

2. There is no doubt that our current understanding of thermodynamic parameters of the DNA double helix and mismatched pairs is pretty solid so the fact that the authors obtained a good agreement between theory and experiment for their artificial system is not surprising and adds very little, if any, to our understanding the DNA thermodynamics.

Technical comment

Although the authors originally formulate the problem under consideration in very general terms, to perform a theoretical treatment they make a number more or less arbitrary assumptions, which allow them to solve the problem under consideration. For example, to calculate melting curves of fully matched duplexes they start with the nearest neighbor approximation, which takes into account the parameters of heterogeneous stacking obtained from melting experiments with long DNA duplexes, rather than using a simplified version of the melting theory, which considers the duplex stability as a function of only the duplex GC-content. But in the end, they de facto arrive at the simplified version anyway. They would make their reasoning more convincing and straightforward if they relied on a finding by Vologodskii and Frank-Kamenetskii (Phys Life Rev 25, 1-21, 2018) that theoretical models, which take into account heterogeneous stacking, do not yield better predictions of oligonucleotide duplex melting temperatures than the simplest model, which relies only on the GC-content.

Reviewer #2: This is a very well-done paper dealing with the ability of complementary DNA strands to find each other among a plethora of other strands, from completely incompatible to partially compatible, and taking into account also imperfect structural pairing.

Experiments and theory nicely go hand-in-hand, and there its much to say, all in all.

Nonetheless, I want to probe the authors in two directions:

1) the theory works too well, given many approximations. It would be good if the authors could add a section for limitations of the analytical model, and where it could go wrong. This would provide a better view for potential users of the framework about when and where this treatment is appropriate and when they should instead be careful

2) Could the authors think about specific mixtures of DNA sequences designed in such a way to maximise complementarity. For example, what about ensuring a certain minimal Hamming distance (or the like) between sequences to reduce mismatches?

I think these would be two valuable additions both in the direction of acknowledging the limitations of the model, and in terms of proposing how to make the model "useful" beyond the present experiments.

Reviewer #3: The paper by Simone De Leo and co-workers present an analysis of the melting of random DNA oligomers in "Superdiverse Environments". The article is rather technical for the journal; however, it includes detailed supplementary information.

It is well done and shows exceptional agreement between the experimental observations and the theoretical predictions.

However, a deeper discussion of the results in the context of modern DNA in living cells, implications of their findings and possible limitation or strength due to the oligomer lengths would have been more commendable for a larger scientific audience. For instance, a more extensive discussion on the PCR primer selectivity and possible errors in the amplification would be interesting in different applications.

**Have the authors made all data and (if applicable) computational code underlying the findings in their manuscript fully available?**

Reviewer #1: Yes

Reviewer #2: **No: **From my reading, I did not see any claim of public data/codes

Reviewer #3: Yes

PLOS authors have the option to publish the peer review history of their article (what does this mean?). If published, this will include your full peer review and any attached files.

Reviewer #1: No

Reviewer #2: No

Reviewer #3: No

Figure Files:

Data Requirements:

Reproducibility:

References:

---

## [Editor Report · Decision Letter 1]

22 Mar 2022

Dear prof Bellini,

We are pleased to inform you that your manuscript 'Pairing Statistics and Melting of Random DNA Oligomers: finding your Partner in Superdiverse Environments' has been provisionally accepted for publication in PLOS Computational Biology.

Best regards,

Eugene I. Shakhnovich

Guest Editor

PLOS Computational Biology

Nir Ben-Tal

Deputy Editor

PLOS Computational Biology

---

## [Editor Report · Acceptance letter]

7 Apr 2022

PCOMPBIOL-D-22-00089R1 

Pairing Statistics and Melting of Random DNA Oligomers: finding your Partner in Superdiverse Environments

Dear Dr Bellini,

I am pleased to inform you that your manuscript has been formally accepted for publication in PLOS Computational Biology. Your manuscript is now with our production department and you will be notified of the publication date in due course.

With kind regards,

Livia Horvath
